Testing the function of dromaeosaurid (Dinosauria, Theropoda) ‘sickle claws’ through musculoskeletal modelling and optimization

Bishop Peter J. pbishop@rvc.ac.uk
Structure and Motion Laboratory, Comparative Biomedical Sciences, Royal Veterinary College , Hatfield , United Kingdom
Geosciences Program, Queensland Museum , Brisbane , Queensland , Australia
Wilson Laura
Electronic publication date: 2019 Aug 28
Publication date: 2019
Volume: 7
Electronic Location ID: e7577
Received 2019 Jun 3; Accepted 2019 Jul 29
Copyright: ©2019 Bishop
Copyright year: 2019
Copyright holder: Bishop
License: This is an open access article distributed under the terms of the Creative Commons Attribution License, which permits unrestricted use, distribution, reproduction and adaptation in any medium and for any purpose provided that it is properly attributed. For attribution, the original author(s), title, publication source (PeerJ) and either DOI or URL of the article must be cited.
License URL: https://creativecommons.org/licenses/by/4.0/

Keywords: Dromaeosaurids, Sickle claws, Musculoskeletal model, Optimization, Biomechanics

Funding: Australian Government Research Training Program Scholarship ERC Horizon 2020 Advanced Investigator Grant 695517 Supported by an Australian Government Research Training Program Scholarship (awarded by Griffith University), and an ERC Horizon 2020 Advanced Investigator Grant (695517, awarded to John Hutchinson). The funders had no role in study design, data collection and analysis, decision to publish, or preparation of the manuscript.

==============================
Dromaeosaurids were a clade of bird-like, carnivorous dinosaurs that are well known for their characteristic morphology of pedal digit II, which bore an enlarged, sickle-shaped claw and permitted an extreme range of flexion–extension. Proposed functions for the claw often revolve around predation, but the exact manner of use varies widely. Musculoskeletal modelling provides an avenue to quantitatively investigate the biomechanics of this enigmatic system, and thereby test different behavioural hypotheses. Here, a musculoskeletal model of the hindlimb and pes of Deinonychus was developed, and mathematical optimization was used to assess the factors that maximize production of force at the claw tip. Optimization revealed that more crouched hindlimb postures (i.e., more flexed knees and ankles) and larger flexor muscle volumes consistently increased claw forces, although the optimal degree of digit flexion or extension depended on assumptions of muscle activity and fibre operating range. Interestingly, the magnitude of force capable of being produced at the claw tip was relatively small, arguing against regular transmission of a large proportion of body weight into a substrate principally via the claw tip. Such transmission would therefore likely have needed to occur via more proximal parts of the foot. Collectively, the results best support a grasping function for digit II (e.g., restraint of prey smaller than the dromaeosaurid’s own body size), although other behaviours involving flexed hindlimbs cannot be excluded.

Introduction

Dromaeosaurids were a long-lived, geographically widespread and highly diverse clade of small- to medium-sized theropod dinosaurs that were very closely related to birds (Norell & Makovicky, 2004; Turner, Makovicky & Norell, 2012; Turner et al., 2007). One of the most striking anatomical features of this group is a highly modified pedal digit II (second toe), the likes of which are not observed in any living species. Specifically, the articular facets of the first and second phalanges enabled extreme digit hyperextension, in addition to strong digit flexion, and the ungual (claw) was larger and more strongly curved than those of the other pedal digits (Fig. 1A; Norell & Makovicky, 1997; Norell & Makovicky, 2004; Ostrom, 1969b; Senter, 2009; Turner, Makovicky & Norell, 2012). This morphology is also present in troodontids, which are often recognized as the sister group to dromaeosaurids (Turner, Makovicky & Norell, 2012), although see Godefroit et al. (2013) for a different interpretation. In phylogenetically derived dromaeosaurids (eudromaeosaurs) such as Deinonychus, Velociraptor and Utahraptor, the ungual of digit II becomes further enlarged and sickle-shaped, in association with relative shortening of the metatarsus (Ostrom, 1976; Turner, Makovicky & Norell, 2012).

Figure 1 The famed ‘sickle claw’ of pedal digit II in dromaeosaurids and its hypothesized uses.

(A) Schematic illustration of the left pes of Deinonychus (after Ostrom, 1969b), which also illustrates the widespread inference that digit II was retracted off the ground when the claw was not in use. (B–H) Various hypotheses previously proposed for how the claw was used in life. (B) Kicking or slashing of prey. (C) Used for gripping onto the flanks of struggling prey. (D) Using body weight to drive the claws down the side of the prey’s flank. (E) Piercing or slashing specific vital areas of the prey. (F) Pinning down and immobilizing prey to be dispatched by the mouth and forelimbs. (G) Intra- or interspecific defence. (H) Digging out prey from nests or burrows. The schematics in B–H are based as closely as possible on the original descriptions (and sometimes illustrations) of the hypothesized behaviours in the literature; see main text for citations. Also note that integument (e.g., feathers) is omitted for clarity.

Owing to its unique morphology and mobility, the function of pedal digit II has featured prevalently in discussions of dromaeosaurid palaeobiology, especially predation. In this context, it is useful to define (relative) prey size categories following the prior framework of Fowler, Freedman & Scannella (2009) and Fowler et al. (2011). Specifically, ‘small’ prey is able to be contained entirely within the foot (encircled by flexed digits and claws); ‘large’ prey is too large to be able to be contained within the foot, but can be held down by the predator’s body weight; ‘very large’ prey is too large to be held down by the predator’s own weight. A wide variety of hypotheses have been proposed for the how digit II was used in life, which include:

1. Used in combination with kicking behaviour of one hindlimb (while balancing on the other limb) to cut, slash or disembowel prey (Fig. 1B; Adams, 1987; Ostrom, 1969b; Ostrom, 1990); the prey animal is often presumed to be of very large body size.

2. After leaping onto the flanks of a very large prey animal, the claws are used to pierce and grip the hide of the prey, allowing the predator to hold onto struggling prey and position itself appropriately for delivering bites (Fig. 1C; Manning et al., 2009; Manning et al., 2006). The efficacy of this technique is supported by analogy with the curved manual claws of felids (Bryant et al., 1996) and was also demonstrated using a life-sized physical model by Manning et al. (2006), although Fowler et al. (2011) raised concerns over the latter study’s model accuracy.

3. After leaping onto the flanks of a prey animal (which again is of very large size), the predator uses its body weight to drive the claws into and down the sides of the prey to inflict large wounds (Fig. 1D; Henderson, 2012). A variant of this is where the predator actively employs a kicking and slashing action whilst on the prey’s flanks (Paul, 1988).

4. The claw is used to pierce or slash specific parts of the prey, targeting vital areas such as the blood vessels or trachea in the neck (Fig. 1E; Carpenter, 1998). This is often inferred to be the case in the famous ‘fighting dinosaurs’ specimen of a Velociraptor preserved alongside a Protoceratops (Barsbold, 2016; Carpenter, 1998; Kielan-Jaworowska & Barsbold, 1972).

5. The claws are used to help pin down and restrain prey of small or large size, allowing it to be attacked and dismembered by the mouth and forelimbs (Fig. 1F; Fowler et al., 2011). The morphology of the claws and feet in many dromaeosaurids is consonant with a gripping or grasping function (Fowler et al., 2011).

6. Used in defence, such as by kicking as often used by various extant birds (Fig. 1G; Colbert & Russell, 1969; Senter, 2009), although whether this occurred during inter- or intraspecific interactions has not been specified.

7. Used in a digging action to extract small prey from nests or burrows, possibly involving a ‘hook and pull’ motion of a strongly flexed digit (Fig. 1H; Colbert & Russell, 1969; Senter, 2009; Simpson et al., 2010).

It is worth noting that the different proposed behaviours involve both markedly different degrees of digit flexion–extension and different whole-limb postures. In contrast to the uncertainty surrounding how digit II was used, however, it is widely agreed that when not in use (e.g., locomotion) it was held in a retracted (extended) state, with the claw held off the ground as illustrated in Fig. 1A. This is supported by numerous didactyl and narrow-gauge bipedal fossil trackways of Mesozoic age, for which dromaeosaurids or troodontids are the only reasonable makers (e.g., Li et al., 2008; Lockley et al., 2016; Xing et al., 2018; Xing et al., 2015; traces of the distal end of digit II are almost always absent). It is also consistent with osteological range of motion (e.g., Colbert & Russell, 1969; Currie & Peng, 1993; Ostrom, 1969b; Paul, 1988; Senter, 2009) and articulated fossil specimens (e.g., Csiki et al., 2010; Norell & Makovicky, 1997; Xu et al., 2003).

One aspect of digit II function in dromaeosaurids that has received little attention is the musculoskeletal mechanism (or mechanisms) that underpinned claw use. In particular, the extreme range of motion that digit II was apparently capable of implies substantial changes in length of the musculotendon units (MTUs) that actuated the claw across the range of flexion–extension. This may have had important consequences for force production capacity, as the amount of force that a muscle can actively develop depends on how stretched or contracted its constituent fibres are Turner et al. (2007) (Fig. 2; McMahon, 1984; Millard et al., 2013; Zajac, 1989). In addition to this, force can also be passively produced, both through stretch of the muscle fibres and connective tissues and stretch of the in-series tendon. Thus, the force–length relationships of muscle and tendon may constrain the kinds of digit and whole-limb postures at which maximal claw force could have been attained, and in turn influence the manner in which the claw was used in life.

Figure 2 Mathematical representation of muscle architecture and its force–length–velocity relationships.

(A) At the organ level, muscle is typically represented with a Hill-type computational model, which comprises a contractile element (CE) in parallel with a passive elastic element (PEE), which are in turn in series with a second elastic element (SEE) representing the tendon; see Zajac (1989) and Millard et al. (2013) for more detail. This model factors in the architectural parameters of optimum fibre length (ℓo), pennation angle at optimum fibre length (αo) and tendon slack length (Ls). (B–D) The relationships between normalized force (F *, equal to force divided by maximum isometric force), normalized fibre length (ℓ*, equal to fibre length divided by ℓo) or normalized tendon length (L *, equal to tendon length divided by Ls) and normalized fibre velocity (v *, equal to fibre velocity divided by maximum contraction velocity). (B) Force–length relationships of the active (red) and passive (blue) muscle components. (C) Force–length relationship of tendon. (D) Force–velocity relationship of the active muscle component; in the current study this is ignored as all analyses are static only. The curves in B–D are based on the formulation of De Groote et al. (2016).

Computational modelling of the musculoskeletal system provides a robust and quantitative means of addressing this question, and in turn provides an avenue to investigating the function of digit II in dromaeosaurids. Previously, musculoskeletal models have shed insight on diverse palaeobiological topics, including muscle leverage (e.g., Bates, Benson & Falkingham, 2012; Bates & Schachner, 2012; Hutchinson et al., 2008; Hutchinson et al., 2005; Maidment et al., 2014), bite forces (e.g., Bates & Falkingham, 2012; Bates & Falkingham, 2018; Lautenschlager et al., 2016), posture and locomotion (e.g., Bates, Benson & Falkingham, 2012; Bishop et al., 2018; Hutchinson et al., 2005; Nagano et al., 2005; Sellers et al., 2017). The present study developed a musculoskeletal model of the dromaeosaurid hindlimb to examine how the force–length relationships of muscle and tendon may influence claw force production. Framing the question as an optimization problem, the aim was to determine the combination of factors (including digit and limb posture) that maximize claw force, with the view to testing the aforementioned hypotheses of how digit II was used in life. In addition to addressing the specific question of claw use in dromaeosaurids, this study also provides a methodological framework that may be adapted and used to address other palaeobiological questions in the future.

Methods

Approach of the current study

Before the specific methodological details are presented in full, the key premises and assumptions of the current study are outlined here.

Optimization

The highly modified osteology of pedal digit II, in concert with the inference that it was not used in locomotion (Li et al., 2008; Lockley et al., 2016; Xing et al., 2018; Xing et al., 2015), suggests that it was used for a very specific purpose in life. Musculotendon anatomy was therefore probably quite highly adapted (‘tuned’) for that single purpose, in contrast to other muscles in the hindlimb which would be required to execute many different activities (see Hutchinson et al., 2015). As such, the question of dromaeosaurid digit II function can be approached via the methods of mathematical optimization. That is, the goal is to relate how claw force varies with musculotendon anatomy, digit flexion and whole-limb posture, to identify the circumstances in which the force produced at the claw tip is maximized.

Statics

Common to each of the proposed hypotheses for dromaeosaurid digit II use (Figs. 1B–1H) is an implicit assumption that the force applied by the claw to the substrate (e.g., prey) is of greater importance than the speed at which the claw tip moves with respect to more proximal limb segments (e.g., metatarsus). As a consequence, all simulations in the present study were static only; this assumption also helped to make the system more tractable for analysis and serves as a useful starting point that may be built upon in the future.

Musculature

During claw use, regardless of the source of the force being applied through the claw (e.g., intrinsic flexor musculature of digit II, more proximal muscles in the hindlimb, body weight, limb inertia), the muscle that flexed the claw itself would have to be capable of matching or exceeding these forces. Otherwise, upon encountering reaction forces from the substrate, the claw (and possibly also more proximal parts of digit II) would undergo relative extension, diminishing its effectiveness. It is likely that multiple muscles (both flexor and extensor) would have actuated digit II in life. The extant phylogenetic bracket (Witmer, 1995) of dromaeosaurids, crocodilians and birds, possess multiple digital flexors and multiple digital extensors in the hindlimb (Allen et al., 2015; Baumel et al., 1993; Cong, 1998; George & Berger, 1966). However, aside from the flexor digitorum longus and extensor digitorum longus, the homology of these muscles among archosaurs remains unclear (Carrano & Hutchinson, 2002; Hutchinson, 2002). Dromaeosaurids were probably capable of controlling flexion–extension of digit II independent of the other digits, as suggested by previously proposed behavioural hypotheses and their inferred ability to retract the digit when it was not in use. Hence, any muscles that actuated digit II would have been separate from those actuating other digits; as the flexor digitorum longus and extensor digitorum longus are the only muscles to attach to the ungual in extant archosaurs, dromaeosaurids may therefore be inferred to have had at the very least separate digit II derivatives of the flexor and extensor digitorum longus. These two muscles would have been the two most important muscles in digit II: if they were not strong enough to actuate the claw appropriately, then the digit’s function would be diminished. As such, in the current study only a single flexor and extensor muscle were modelled, attaching to their respective tubercle on the ungual; these are responsible for actuating the entire digit. This simplifying assumption again helps make the system more tractable and can be built upon in future analyses.

Digit II retraction

As noted above, dromaeosaurids are inferred to have held digit II in a retracted state when it was not in use. The mechanism by which this was achieved may have an important influence on musculotendon behaviour when the claw was in use, and so is deserved of careful consideration. There are at least four possible ways in which digit II was retracted above the ground:

1. Snap ligaments that spanned the metatarsophalangeal and interphalangeal joints, which were engaged when the digit was retracted beyond a certain point (effected by active contraction of extensor musculature), upon which the digit would be passively held in place (Manning et al., 2006). This is analogous to the retractable claws of felids (Bryant et al., 1996; Gonyea & Ashworth, 1975).

2. In a similar fashion to the digital flexor tendons of many extant neognath birds, the extensor musculature of digit II possessed a ‘tendon locking mechanism’ (Einoder & Richardson, 2006; Einoder & Richardson, 2007; Quinn & Baumel, 1990) that involved a ratchet-like interaction between the tendon(s) and the surrounding sheath(s). This is functionally similar to snap ligaments, in that active muscle contraction engages the mechanism, upon which the digit is passively held in a retracted state. (See also Manning et al., 2009, who considered the possibility of a tendon locking mechanism in the flexor tendons that may have assisted with prey apprehension.)

3. The extensor musculature was constantly active, producing force that continually held the digit in a retracted state. When the musculature ceased firing, the digit would flex under its own weight (and possibly due to passive elasticity in stretched flexor musculature) back to an unretracted state.

4. The normal resting lengths of the flexor and extensor muscles were such that the digit was elevated above the ground when the muscles were inactive. That is, the weight of the digit (causing the digit to flex downward) was counteracted by passive elasticity in the extensor muscle and tendon (causing the digit to extend upward). This is functionally analogous to ‘tenodesis grasp’ in humans, where passive flexing of the digits occurs when the wrist undergoes extension (Mateo et al., 2013).

Each of the above mechanisms is speculative to some degree. There is no osteological evidence of snap ligaments in the pedes of non-avian theropods, nor are they known in the pedes of extant crocodilians or birds. Osteological evidence is also lacking for a tendon locking mechanism (although admittedly this would be difficult to detect, even in extant taxa), and among extant archosaurs it is only known among neognath birds. Hence, inferring the existence of mechanisms 1 or 2 in dromaeosaurids is highly speculative; at best, a level II′ inference in the scheme of Witmer (1995). Mechanisms 3 and 4 are less speculative in that they do not require any specialized anatomical adaptations; of the two, mechanism 4 is the less energetically demanding, being entirely passive in nature. It is therefore deemed that mechanism 4 provides the most plausible strategy to maintaining digit II in a retracted state when not in use; this forms the basis for a constraint implemented in the optimizations (below).

Musculoskeletal modelling

A three-dimensional (3-D) musculoskeletal model of the hindlimb and pes of Deinonychus antirrhopus was used as the basis of the current study (Fig. 3). Deinonychus, which means ‘terrible claw’ (Ostrom, 1969a), was a relatively large (up to 3 m and 170 kg; Turner et al., 2007) dromaeosaurid that lived in the Early Cretaceous of North America, and has been frequently studied with respect to dromaeosaurid palaeobiology (e.g., Fowler et al., 2011; Gignac et al., 2010; Gishlick, 2001; Ostrom, 1969b; Ostrom, 1994). The geometry of the pes skeleton was acquired through X-ray computed tomographic scanning (Toshiba Aquilion 64, 135 kV peak tube voltage, 250 mAs exposure, 750 ms exposure time, 0.5 mm slice thickness, 0.625 mm pixel resolution) of a complete articulated pes, specimen MOR 747 (Museum of the Rockies, Bozeman, MT, USA). The resulting scans were segmented using Mimics 17.0 (Materialize NV, Leuven, Belgium) to produce surface meshes, which were then refined in 3-matic 9.0 (Materialize NV, Leuven, Belgium) and ReMESH 2.1 (Attene & Falcidieno, 2006). More proximal limb bones were sculpted digitally using Rhinoceros 4.0 (McNeel, Seattle, USA) based on comparison to the literature (e.g., Ostrom, 1969b; Ostrom, 1976), and were scaled appropriately with respect to the pes.

Figure 3 Musculoskeletal model of the right hindlimb of Deinonychus used in the study.

(A) The full model in medial (left) and anterior (right) view, showing the wrapping surfaces (blue) used to help constrain MTU paths (red). (B) Three whole-limb postures were tested for, spanning from extended (1) to crouched (3) configurations. (C) Two different lengths of each MTU were tested for, short MTUs originating from the proximal tibia (left) and long MTUs originating from the distal femur (right). (D) The motion of the MTP joint (and the two IP joints that were programmatically coupled to it) ranged from −65° of flexion through to 60° of extension. Arrows show the points at which MTU length distal to the ankle was measured, used in the calculation of bounds to tendon slack length (see Table 1). For scale, the femur in the model is 291 mm long.

The musculoskeletal model was constructed in NMSBuilder (Martelli et al., 2011; Valente et al., 2014) for use in OpenSim 3.3 (Delp et al., 2007). All joints were assigned a single degree of freedom only, with the rotation axis fixed with respect to the ‘parent’ body in all cases. As the proximal limb bone geometries were sculpted, the knee and ankle joint axes were a priori set as parallel to the global y-axis (mediolateral axis). The axes of the first and second interphalangeal (IP) joints of digit II were determined in 3-matic, by fitting a cylinder to the outer margins of the ginglymoid parts of the articular surfaces of the proximal bone involved, with the axis of the cylinder taken to be the axis of rotation. Due to the strong asymmetry of the metatarsophalangeal (MTP) joint of digit II, motion at this joint was modelled with a helical axis, with coupled rotation about and translation along the axis. The location, orientation and amount of translation per unit rotation of the helical axis was determined with the KineMat toolbox (Reinschmidt & Van den Bogert, 1997; see also Spoor & Veldpaus, 1980) for MATLAB 9.5 (MathWorks, Natick, USA), using three landmarks located on phalanx II-1 at pre-determined positions of maximum flexion and extension with respect to metatarsal II. Subsequent visual inspection of the helical motion in OpenSim indicated that the resulting movement was satisfactory, keeping the distance between opposing articular surfaces approximately constant across the area of articulation. The ranges of motion assigned to the MTP and IP joints were based on preserved articular surface geometry, and included the range hypothesized by previous studies (e.g., Ostrom, 1969b; Senter, 2009). So as to simplify the system for analysis, the motions of the two IP joints were programmatically coupled to that of the MTP joint in the model. That is, the angles of the two IP joints were functions of the MTP joint angle, such that only a single degree of freedom (MTP joint angle) was required to describe all three joints that effected digit flexion or extension. The coupling of motions between joints was linear, and was defined by the points of maximum flexion and extension for each joint: (1) θIP1=1.04211×θMTP+39.75756

(2) θIP2=0.69474×θMTP−3.49496,

where θIP1, θIP2 and θMTP are the first IP, second IP and MTP joint angles, respectively, measured in degrees. Ultimately, this meant that the digit was capable of 125° of motion (−65° flexion to 60° extension; Fig. 3D).

Table 1 Parameters defining the different variations in the model tested, as well as the bounds on the design variables during the optimization.

Postural angles are measured relative to the model’s ‘neutral posture’, which is where the limb is fully straightened. Note that thigh angle is only used to reorient the model with respect to the global coordinate system.

Parameter	Posture 1	Posture 2	Posture 3	
Thigh angle (°)	−15	−40	−65	
Knee angle (°)	−25	−70	−115	
Ankle angle (°)	20	70	120	
Threshold MTP angle (°)	35	25	15	
Threshold moment (Nm)	−0.02529	−0.02823	−0.02866	
ℓo lower bounds (m)	0.02	
ℓo upper bounds (m)	0.65	
Ls lower bounds (m)	0.191 (extensor), 0.237 (flexor)	
Ls upper bounds (m)	Long MTU: 0.423 (extensor), 0.471 (flexor)	
Short MTU: 0.311 (extensor), 0.369 (flexor)	
αo lower bounds (°)	0a	
αo upper bounds (°)	60a	
Notes.

a 20–35° in sensitivity analysis.

As noted above, only a single flexor and a single extensor muscle were modelled, which are inferred to be derivatives of the flexor digitorum longus and extensor digitorum longus, respectively. These muscles exhibit disparate origins among extant archosaurs: in crocodylians they both originate from the distal femur, whereas in birds they originate from the proximal tibia (Allen et al., 2015; Baumel et al., 1993; Cong, 1998; George & Berger, 1966). As such, two geometric variants of both muscles’ MTU actuators were used: a ‘short’ version that originated from the proximal tibia, and a ‘long’ version that originated from the distal femur (Fig. 3C). The 3-D course of the MTUs were constrained to follow anatomically realistic paths across the model’s entire range of motion, using a combination of via points and cylindrical or toroidal wrapping surfaces (Fig. 3A; Delp et al., 1990; Garner & Pandy, 2000); the courses of the short and long MTUs were identical from the distal tibia onwards towards the ungual.

Insofar as the MTUs were concerned, the musculoskeletal model was used only to provide information on geometrical relationships, namely, how MTU lengths and moment arms varied with respect to joint angle. The actual modelling of MTU activation–contraction dynamics was undertaken in MATLAB, using the formulation of De Groote et al. (2016). This formulation is a Hill-type model (see Fig. 2A) that uses an implicit representation of activation–contraction dynamics with tendon force as a state variable, and which is computationally robust and conducive to efficient numerical optimization (e.g., by enabling algorithmic differentiation). The musculoskeletal model was also used to calculate the moment about the MTP joint due to the weight of the digit at a given MTP angle (used in the constraints described below). Mass properties of each digit segment were defined by first modelling the soft tissues with basic geometries (elliptical frusta or cylinders) in Rhinoceros, and then assigning a bulk density to each segment (1,000 kg/m3 for phalanges II-1 and II-2; 1,500 kg/m3 for the ungual).

Optimization

For simplicity, all optimization simulations in the present analysis were static only. The musculoskeletal model was used to identify the circumstances in which the force produced at the claw tip is maximized. The optimization problem was solved for each MTP angle and was posed thus: Given

(3) Fflex=fℓO,flex,LS,flex,αO,flex,Fmax,flex,aflex,lflex

(4) Fext=fℓO,ext,LS,ext,αO,ext,Fmax,ext,aext,lext

(5) lflex=fθMTP,posture

(6) lext=fθMTP,posture

(7) rflex,MTP=fθMTP

(8) rext,MTP=fθMTP

(9) rflex,IP2=fθMTP

(10) rext,IP2=fθMTP

Optimize

(11) maxFclaw=Fflexa=1⋅rflex,IP2+Fexta=0⋅rext,IP2rclaw

Subject to

(12) ℓflex,θMTP=−65°∗,a=1≥0.5

(13) ℓflex,θMTP=60°∗,a=1≤1.5

(14) ℓext,θMTP=−65°∗,a=1≤1.5

(15) ℓext,θMTP=60°∗,a=1≥0.5

(16) Fflexa=0⋅r flexθMTP=thresh+F exta=0⋅r extθMTP=thresh≥M weightθMTP=thresh.

Equations (3) and (4) denote the models of muscle activation–contraction dynamics; they relate MTU force (F) to optimum fibre length (ℓo), tendon resting or slack length (Ls), pennation angle at optimum fibre length (αo), maximum isometric force (Fmax), activation (a) and MTU length (l), with the subscripts of ‘flex’ or ‘ext’ referring to flexor and extensor MTUs, respectively. Equations (5)–(10) describe the geometric aspects of the system as represented by the musculoskeletal model. Equations (5) and (6) describe how MTU length varies as a function of MTP angle (θMTP) and whole-limb posture, whilst Eqs. (7)–(10) describe how MTU moment arms about a given joint (MTP or second IP) vary as a function of MTP angle.

Equation (11) is the objective function which is to be maximized, and describes moment balance about the second IP joint. The flexor MTU is maximally active (a = 1), whereas the extensor MTU is quiescent (a = 0) and can only exert force passively, and rclaw is the perpendicular distance from the axis of rotation of the IP2 joint to the claw tip, which was determined in Rhinoceros to be 76 mm.

Equations (12)–(16) denote constraints that must be satisfied. Equations (12)–(15) stipulate that the normalized fibre lengths of both flexor and extensor MTUs must remain within reasonable operating ranges at maximal activation, lest they are capable of producing little force (Fig. 2B). The default range assigned here was 0.5 ≤ ℓ* ≤ 1.5 (Rankin, Rubenson & Hutchinson, 2016; but see below). Equation (16) describes the tenodesis-like mechanism that was identified above as probably responsible for holding digit II clear of the ground, and stipulates that the musculoskeletal anatomy that optimizes claw function must also be compatible with holding digit II off the ground. It states that, at some threshold angle (θMTP = thresh), the sum of the moments of the flexor and extensor MTUs about the MTP joint (produced only through their passive components, i.e., a = 0), equals or exceeds the moment due to weight of the digit (Mweight). If this is true, then the digit will be able to be held aloft at that threshold angle, or at even higher extension angles.

A total of six design variables were optimized: fibre length, tendon slack length and pennation angle, for both flexor and extensor MTUs. During each iteration, the maximum isometric force of the MTUs was recalculated accordingly as

(17) Fmax=V⋅σ⋅cosαOℓO,

where V is muscle belly volume and σ is maximum isometric stress, for which a value of 300,000 N/m2 was initially used (e.g., Bates & Falkingham, 2012; Hutchinson, 2004a; Sellers et al., 2013). Muscle volume was initially set at 25 cm3 for the flexor and 12.5 cm3 for the extensor, as a subjective guess based on the dimensions of the musculoskeletal model. The permissible values for the design variables during the optimization were constrained between lower and upper bounds that were defined on the basis of the geometry of the musculoskeletal model (Table 1). The lower bound for tendon slack length was set as the minimum length achievable by the respective MTU distal to the ankle across all postures (Fig. 3D). The upper bound for tendon slack length was set at 90% of the minimum length achievable by the MTU across all postures, and varied depending on the MTU length variant used. An overly generous range of permissible values for pennation angle was used (0–60°), which is considerably broader than the range observed in the digital flexor and extensor muscles of extant archosaurs (typically 20–35°; Allen et al., 2010; Allen et al., 2015; Hutchinson et al., 2015; Lamas, Main & Hutchinson, 2014; Paxton et al., 2010; Smith et al., 2006). Given that the flexor and extensor muscles of digit II in dromaeosaurids are presumed to have been ‘tuned’ for executing a single, non-locomotor behaviour, they may potentially have had a markedly different architecture compared to their homologues in extant archosaurs, which are required to execute a variety of different behaviours (locomotor and non-locomotor; see Hutchinson et al., 2015). It was therefore not considered justified in this case to constrain pennation angles to the same range of values as observed in extant archosaurs –the ‘everyanimal’ issue of Pagel (1991).

The optimization was implemented in a set of custom MATLAB scripts that used CasADi 3.4.5 (Andersson et al., 2019), a suite of tools for nonlinear optimization and algorithmic differentiation. The relationships between MTU moment arms or lengths and joint angles were derived from the musculoskeletal model and fed directly into the optimization framework, obviating the need to interface with OpenSim, which speeds up computation (e.g., by facilitating algorithmic differentiation) and avoids discontinuities in the optimization problem. All other aspects of the system were implemented directly into the scripts. The optimization used the open-source solver IPOPT 3.12.3 (Wächter & Biegler, 2006), accessed via the CasADi interface.

Sensitivity analysis

Optimizations were run for a variety of different combinations, to both test different limb postures and assess sensitivity to unknowns in the musculoskeletal model. Three whole-limb postures were tested (Fig. 3B, Table 1), spanning extended through to crouched limb configurations; each of these postures necessitated a different value for the threshold MTP angle in the constraint described by Eq. (16) (Table 1). Two different lengths of both flexor and extensor MTUs were tested, on account of uncertainty in muscle origin (see above; Fig. 3C). Lastly, two additional variants in flexor and extensor muscle belly volume were tested for (±25% of initial value). A total of 108 different combinations of variants were subject to optimization, which took approximately 8.5 h to solve using a computer with a 2.4 GHz processor. In addition, the sensitivity of optimization results was tested for with regards to four further model assumptions. Firstly, normalized fibre lengths of flexor and extensor MTUs were constrained to operate within the more restricted (optimal) range of 0.75 ≤ℓ* ≤ 1.25 (affecting Eqs. (12)–(15)). Secondly, when digit II was not in use, it was now presumed to be held in a retracted state through active contraction of the extensor MTU (a = 0.5, affecting Eq. (16)). Thirdly, muscle strength was increased by doubling maximum isometric stress to 600,000 N/m2 (affecting Eq. (17)). Lastly, the upper and lower bounds for pennation angle of both flexor and extensor muscles were restricted to 20–35°, comparable to the range observed in the digital flexor and extensor muscles of extant archosaurs.

Results

Initial system configuration

Using the initial set of assumptions about the system, the maximal claw tip force that is possible at each MTP angle is shown in Fig. 4, summarizing all 108 combinations of whole-limb posture, muscle bulk and muscle length. The mean and range shown for a given posture in Fig. 4A is based on the curves for 36 combinations. These curves (and their derived mean and range curves) are not a profile of force versus MTP angle for a given combination of MTU parameters; rather, each curve is the maximal envelope of all force–angle curves for a given combination, as shown in Fig. 4B. That is, in Fig. 4B there are 126 curves, representing the force–angle profile that results from the optimal combination of MTU parameters identified for each degree of MTP flexion–extension (−65° to 60°); the maximal envelope of these 126 curves contributes one curve to the data summarized in Fig. 4A. Across the 108 combinations tested, claw force is maximized at MTP angles of between −15 to 50°, with a global maximum of 18.9 N achieved at an angle of 20°, corresponding to modest digit extension. The results are parsed by whole-limb posture in Fig. 4A, by muscle volume in Figs. 4C and 4D, and by muscle length in Figs. 4E and 4F. The magnitude of claw force is markedly variable depending on limb posture, muscle bulk and muscle length, and likewise the MTP angle at which maximum claw force is achieved also varies considerably with respect to these factors. Nevertheless, two clear trends are that more crouched postures (Fig. 4A) and a larger flexor muscle (Fig. 4C) consistently produce higher claw forces.

Figure 4 Optimization results for the initial system configuration, showing claw force (Fclaw) plotted against metatarsophalangeal joint angle (θMTP).

(A) Results for the 108 combinations of posture, muscle volume and muscle length, parsed by posture. (B) Results for claw force for one of the 108 combinations include 126 force–angle curves calculated for the optimal musculotendon parameters identified for each degree of the range of θMTP; the maximal envelope of these curves (dotted line) contributes data to panel A (arrow). (C–F) Results for the 108 combinations parsed by flexor muscle volume (C), extensor muscle volume (D), flexor muscle length (E) and extensor muscle length (F). In A, C–F, curves show the mean values, shaded regions denote total range, and crosses denote maxima for each of the 108 curves that are represented in the plots.

Optimal MTU parameters

For each of the 108 combinations tested, the resulting optimal values for the six design variables at each MTP angle are shown in Fig. 5, parsed by posture (see also Figs. S1–S4). The optimal fibre length (ℓo) and tendon slack length (Ls) of the flexor muscle vary largely in tandem with each other across much of the range of MTP angles, as a consequence of the optimal pennation angle (αo) being 0° (i.e., parallel fibred) across this range, particularly for the ‘long flexor’ combinations (Fig. S3E). Low or zero αo, whilst it decreases the magnitude of maximum muscle force (Eq. (17)), enables the muscle fibres to be longer, allowing them to operate closer to the peak in their force–length curve across a larger range of MTU length change. The results here (i.e., the tendency to retrieve the lowest αo possible) therefore directly stem from the large range of flexion–extension that digit II is capable of. The manner of variation in the optimal values of ℓo, Ls and αo for the extensor muscle is less straightforward, which is possibly due to it having to satisfy more requirements in a passive state (activation = 0). Notably, extensor Ls tended toward the lower bounds for allowable values over much of the range of MTP angles, which may be related to the requirement for passive tendon stretch contributing toward holding digit II off the ground; a lower Ls produces more passive force for the same absolute amount of stretch. Additionally, across at least part of the range of MTP angles, the optimal value for extensor αo reached the lower or upper bounds of the permissible range of angles. As with claw force, the optimal values of the MTU parameters can vary considerably: for a given MTP angle, they can vary depending on limb posture, muscle volume and muscle length, and for a given combination of limb posture, muscle volume and muscle length, they can vary depending on the MTP angle. In addition to posture having a pronounced effect on the optimal MTU values (Fig. 5), muscle length also has a distinct influence on the optimal values retrieved (Figs. S3–S4), with flexor length affecting flexor MTU parameters, and vice versa for the extensor, but little ‘cross muscle’ effects.

Figure 5 Optimal values of musculotendon parameters for the initial system configuration, plotted against metatarsophalangeal joint angle (θMTP) and parsed by posture.

(A, B) Optimal fibre length. (C, D) Tendon slack length. (E, F) Pennation angle. A, C and E are for the flexor muscle; B, D and F are for the extensor muscle. Black dashed lines denote minimum, maximum and mean curves across all combinations.

Figure 6 Optimization results for the four sensitivity analyses, showing claw force (Fclaw) plotted against metatarsophalangeal joint angle (θMTP).

(A) More constrained operating range for muscle fibres. (B) Extensor muscle actively contracting in holding digit II off the ground. (C) Muscle strength is doubled. (D) More restrictive bounds on the allowable range of pennation angles in the optimization. All results are parsed by posture. Curves show the mean values, shaded regions denote total range, and crosses denote maxima for each of the 108 curves that are represented in the plots.

Sensitivity analysis

Paralleling the results for the initial system configuration (Fig. 4), in all sensitivity tests conducted it was found that more crouched postures consistently led to higher claw forces being achieved, especially for more flexed digit postures (Fig. 6). Compared to the results for the initial configuration, constraining normalized fibre lengths to a more restricted operating range produced a very different pattern for optimal claw force versus MTP angle (Fig. 6A). In addition to reducing the magnitude of maximal claw force (−32.1 ± 3.5%; mean ± s.d.), this resulted in the maximum force being consistently achieved at a considerably flexed digit posture (MTP angle of −37 to −35°). When the extensor muscle was assumed to be actively contracting to hold digit II off the ground when not in use, this also led to a markedly different pattern in the force–angle curves (Fig. 6B), with maximum claw force achieved at modestly flexed through to extended digit postures (MTP angle of −20 to 15°). Additionally, the magnitude of maximum claw force was higher (30.4 ± 6.6%) compared to the initial results. Doubling muscle strength had very little effect on the results, beyond roughly doubling the magnitude of claw force (90.1 ± 2.9% increase), which is not surprising (Fig. 6C). Lastly, restricting the bounds on allowable values for αo during the optimization had little qualitative effect on the results, although claw force magnitudes were slightly reduced (−11.4 ± 5.1%; Fig. 6D). Mimicking the result noted above for the initial system configuration, the optimal value for flexor muscle αo tended towards the lower bounds of allowable values across much of the range of MTP angles, especially for more crouched postures (Fig. S5A); likewise, the optimal value for extensor αo again reached the lower or upper bounds across at least part of the range of MTP angles (Fig. S5B). This consistent tendency for the optimizer to push against the bounds for αo justifies the use of a broader range of permissible values in the initial system configuration, beyond the restricted range observed in extant archosaurs. Indeed, in the absence of further physical constraints, these results suggest that flexor and extensor muscle αo in dromaeosaurids may have evolved to become markedly outside of the typical range observed in the digital flexor and extensor muscles of extant archosaurs.

Discussion

This study used musculoskeletal modelling, framed within the context of mathematical optimization, to investigate the factors that maximize pedal digit II claw force in dromaeosaurids, using Deinonychus as a case study. The results in turn can help test between the various hypotheses proposed for how the digit and its claw were used in life. Under the initial set of assumptions about the system, claw force was maximized in a crouched whole-limb posture (posture 3), with a modest (20°) level of digit extension (Fig. 4A). However, different assumptions about the behaviour of the MTUs involved sometimes led to claw force being maximized at more flexed digit postures (Fig. 6).

MTU parameters and behaviour

Echoing the sentiments of previous studies (e.g., Bates & Falkingham, 2018; Bates et al., 2010; Hutchinson, 2004b), the results of this study demonstrate the important influence that MTU parameters can have on the biomechanical performance of a system. For a given MTP joint angle, the optimal values for the MTU parameters sometimes varied considerably, depending on the assumed combination of limb posture, muscle bulk and muscle length (Fig. 5, Figs. S1–S4). Moreover, for a given combination, the optimal MTU parameters often varied depending on the MTP angle under consideration (Fig. 5, Figs. S1–S4).The ability to better constrain the range of plausible values for each MTU parameter will help refine the results obtained here, which remains a key challenge in palaeobiological inquiry.

Yet, rather than taking a nihilistic view, musculoskeletal modelling provides a unique perspective to addressing these challenges. It can identify those aspects of the system to which the results are most sensitive, and therefore where future research effort should be directed in order to refine understanding further. For instance, muscle bulk (Figs. S1and S2) and length (Figs. 3 and 4) both exert marked influence on the optimal value for MTU parameters and in turn claw force production; better constraining muscle sizes and origins will help address this. Perhaps the single most influential aspect identified here was the allowable operating range of the muscle fibres: regardless of every other variant in the system, a more constrained operating range (0.75 ≤ ℓ* ≤ 1.25) resulted in claw force being maximized in a very consistent manner, at moderate levels of digit flexion (compare Fig. 6A with Fig. 4A). Underpinning the approach of the present study is the notion that the system under consideration was highly adapted for a single purpose in life. It is therefore conceivable that muscle fibre lengths were so highly optimized that the fibres were indeed able to operate close to the peak in their force–length curve across the whole range of digit II motion. Little is known about the normal operating ranges of muscle fibres in non-mammalian species during in vivo activity, but future experimental study of analogous extant animal systems (e.g., pedal flexor muscles in birds of prey during prey grasping (Sustaita, 2008)) can help validate or refine assumptions made here.

Implications for digit II and claw use

The results obtained here provide new bearing on the validity of different proposed hypotheses of dromaeosaurid digit II and claw use (Table 2). Regardless of MTU input parameters and assumptions about MTU behaviour, it was consistently found that more crouched whole-limb postures lead to higher claw forces being produced. This favours behavioural hypotheses that involve more flexed limb angles (such as grasping or restraining small to large prey at close quarters; Fig. 1F), and detracts from hypotheses involving an outstretched or strongly extended limb (such as slash kicking very large prey at a distance; Figs. 1B and 1G). It is also consonant with the ‘fighting dinosaurs’ specimen, which purportedly shows life behaviours frozen in time (Barsbold, 2016; Kielan-Jaworowska & Barsbold, 1972); the left hindlimb of the Velociraptor is striking its putative prey (the similarly-sized Protoceratops) in a non-extended leg posture, closest to posture 2 that was tested here (cf. Fig. 1E). The hypothesis of prey restraint in particular receives further support if muscle fibres were highly optimized to operate closer to the peak in their force–length curve (see above); here, claw force is maximized at flexed digit II postures, conducive to grasping and holding small prey.

An additional result obtained here is an estimate of the magnitude of the maximum capable claw force. In the initial set of assumptions of the system, the global maximum in claw force across the 108 combinations tested was 18.9 N (Fig. 4A). This is a relatively small quantity, less than a quarter of the flexor muscle force used to produce it in that particular combination (82.0 N). It is also quite small (<5%) in relation to the estimated body weight of the animal the model is based upon (scaling by model femur length to the data of Turner et al. (2007), weight is estimated at 490 N). This is in stark contrast to the forces applied in the physical model of Manning et al. (2006),which equated to 2.5 times body weight. These comparisons do not change substantially under other variations in model assumptions (Fig. 6). That claw force is relatively low is not surprising, because the claw’s large size produces a large ratio of out-lever to in-lever distances; in the model, the distance from the claw tip to the second IP joint is 3.2–4.2 times the moment arm of the flexor muscle about the joint. In life, the claw would have been sheathed in a keratinous covering, extending the tip even further from the second IP joint and further reducing the force produced at the tip. Larger forces would be achievable away from the claw tip nearer the base (smaller out-lever arm), which would help improve the grasping of objects comparable in size to, or smaller than, the radius of curvature of the claw (i.e., small prey). That only low forces were achievable at the tip suggests that the claw by itself was probably not used to support or transmit a large proportion of the animal’s body weight into prey or other substrates (cf. Figs. 1C and 1D). However, body weight could still be used to help restrain prey (small or large) if transmitted via the proximal toes and base of the metatarsus, such as by standing on top of the prey animal (cf. Fig. 1F; Fowler et al., 2011). In addition, the claws could still aid in the maintenance of position and balance by providing grip, for prey of any size.

Table 2 Assessment of how well supported each behavioural hypothesis is by the results of the present study.

As hypotheses have not been previously described in an explicitly quantitative manner, they are codified here in a qualitative fashion only.

Hypothesis	Whole-limb posture	Digit posture	Force required at claw tip	Supported?	
Slash-kicking prey (Fig. 1B)	Extended	Extended	High	No	
Prey riding (Fig. 1C)	Extended to semi-flexed	Moderately extended	High	No	
Prey-mounted flank attack (Fig. 1D)	Extended to semi-flexed	Moderately extended	High	No	
Targeting prey’s vital areas (Fig. 1E)	Semi-flexed to flexed	Variable	Variable (depending on substrate)	Partly	
Prey restraint (Fig. 1F)	Flexed	Flexed	Low	Yes	
Kicking defence (Fig. 1G)	Extended	Extended	High	No	
Digging out prey (Fig. 1H)	Extended to semi-flexed	Flexed	Variable (depending on substrate)	Partly	

Caveats

As in all modelling efforts, simplifying assumptions were made to facilitate more tractable analysis of the system. Possibly the most important is that only two muscles were modelled, yet it is likely that digit II in dromaeosaurids was actuated by multiple flexor and extensor muscles, as in extant archosaurs (Allen et al., 2015; Baumel et al., 1993; Cong, 1998; George & Berger, 1966). Moreover, the actions of multi-joint muscles, such as those modelled here, can be modulated by the actions of other muscles that actuate other parts of the limb (Kuo, 2001); this would be particularly pertinent for the ‘long’ variants of the muscles modelled here, which crossed both knee and ankle joints. The movements of the IP and MTP joints were coupled in the present study, but they may have been capable of some independent movement in life, requiring additional muscles to actuate them. Another important caveat is that the model was analysed as a static system only, yet dynamic effects may be important in life. These include muscle force–velocity effects, tendon recoil and viscous effects from soft tissues that the claw is engaging. Additionally, dynamic effects due to limb segment inertia and movement (especially that of the proximal limb) could be quite important during high-velocity movements, at which point the static assumption used here may become untenable. These assumptions can be tested and built upon in the future, further refining interpretations of digit II function.

Other theropods

It is possible that the function of digit II and its claw may have varied among different dromaeosaurids. The characteristic morphology of digit II was present in phylogenetically basal (and much smaller) taxa that possessed markedly more elongate and gracile limbs than Deinonychus (Turner et al., 2007). This may have resulted in a smaller relative change in length in the MTUs that actuated the digit, potentially influencing force production capabilities. At least one dromaeosaurid (Adasaurus) has an apomorphically reduced claw, yet it retains the characteristic ginglymoid and hyperextensible articular surface morphologies of the penultimate phalanges (Turner, Makovicky & Norell, 2012). Outside dromaeosaurids, the digit II claw of the troodontid Borogovia has reduced curvature, comparable to the relatively straight claws of the other pedal digits, with an apparently reduced capacity for interphalangeal flexion–extension (Osmólska, 1987). Although beyond the scope of this study, modelling of other taxa can help assess if digit II was functionally conserved across dromaeosaurids and troodontids. More broadly, other clades of carnivorous non-avian theropod also developed hypertrophied claws, such as megaraptorans (Hocknull et al., 2009) and spinosaurids (Charig & Milner, 1986), although in these cases it is the claw of manual digit I. Musculoskeletal modelling of the forelimb of these taxa can address the function of their claws, and may in turn shed light on why hypertrophied claws evolved on the manus in these groups, but on the pes in dromaeosaurids and troodontids.

Conclusion

Musculoskeletal modelling, posed within an optimization framework, has provided new insight on the function of pedal digit II and its claw in dromaeosaurids, arguably one of the most enigmatic anatomies among non-avian theropod dinosaurs. Across a variety of soft tissue and postural combinations and assumptions, it was found that more crouched hindlimb postures consistently increased claw force production. Combined with the relatively small magnitude of claw force and considerations of lever mechanics, this lends support to the hypothesis that digit II was employed in the grasping or restraint of prey smaller than the predator (i.e., small to large body size sensu Fowler, Freedman & Scannella, 2009; Fowler et al., 2011). Other behaviours that involve a semi-crouched hindlimb posture are also plausible, such as stabbing or cutting prey at close quarters. In contrast, hypotheses involving strongly extended limbs, or the transmission of a large proportion of body weight into a substrate principally via the claws, are not well supported here. These findings should nevertheless be viewed with circumspect caution, as variation in muscle parameters or assumptions about muscle behaviour can exert an important influence on model results. Importantly, modelling has identified which aspects of dromaeosaurid functional anatomy require further investigation to refine the interpretations made herein.

Supplemental Information

Figures S1–S5 Figures S1-S5

Click here for additional data file.

File S1 Supplementary code and model files

See the included README document for an overview of the files.

Click here for additional data file.

Special thanks to J. Scannella and J. Horner (Museum of the Rockies) for access to the fossil material used in this study, and to K. Ugrin and D. Van Why (Bozeman Deaconess Hospital) for performing the CT scanning. Thanks also to D. Saxby, C. Pizzolato, S. Hocknull, J. Hutchinson, C. Clemente, F. De Groote and A. Falisse for helpful discussion or for assistance with the methodology used here, as well as colleagues of the Structure and Motion Laboratory. The constructive comments of D. Fowler, J. Hutchinson, C. Clemente and an anonymous reviewer on earlier versions of the manuscript are also greatly appreciated.

Additional Information and Declarations

Competing Interests

Author Contributions

Data Availability

The author declares there are no competing interests.

Peter J. Bishop conceived and designed the experiments, performed the experiments, analyzed the data, contributed reagents/materials/analysis tools, prepared figures and/or tables, authored or reviewed drafts of the paper, approved the final draft.

The following information was supplied regarding data availability:

Copies of the computed tomographic scan data are held by the Museum of the Rockies and the Queensland Museum, Brisbane. They can be freely accessed by contacting the Curator of Paleontology at the Museum of the Rockies (john.scannella@montana.edu).

All OpenSim model files and MATLAB code used in the present study are available in the Supplementary Material. See the included README document for details on how to use the files (e.g., required software).

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
