# Peer review of "Testing the function of dromaeosaurid (Dinosauria, Theropoda) ‘sickle claws’ through musculoskeletal modelling and optimization"

_PeerJ, doi:10.7717/peerj.7577_

## Round 0.1 · original submission · Major Revisions

Both reviewers have commented highly positively that your study will represent a valuable addition to the literature and that it is potentially suitable for publication in PeerJ following moderate revision. Reviewer #1 raises the issue that a clear definition of prey size is needed to fully interpret and contextualize the results as presented, and further for comparison with earlier work. Similarly, Reviewer #2 has raised a few substantial comments that require careful attention, particularly relating to justification of aspects of the methods. I also draw your attention to the request to provide .csv files in addition to Matlab files.

·

Basic reporting

no comment

Experimental design

no comment

Validity of the findings

no comment

Additional comments

REVIEW
My identity: Denver Fowler

This manuscript presents a musculoskeletal model of the hindlimbs of dromaeosaurid dinosaurs, seeking to address the question as to the function of the hypertrophied claw on digit II. The model showed that the highest forces occurred during crouched postures, rather than the extended position of the hindlimb required for some suggested behaviors. The study concludes that the D-II ungual is not well suited for exerting much force, and is more supportive of a grasping function.

The manuscript is well written and structured. I appreciate the author attempting to directly address some of the functional hypotheses using modeling of the hindlimb.

My comments concern only content, mainly related to definitions of prey bodysize (lack thereof in this MS), and characterisation of how body weight of the predator is transferred on to its victim when the predator is standing on top of prey. Other than these two small issues the manuscript is good. I would suggest acceptance with minor revisions.

COMMENTS (also see annotated word doc)
I don't think the abstract is clear enough about the difference between body weight for prey restraint (Fig 1. F; ie. Fowler et al, 2011 large and very large prey), and the suggestion that body weight is transferred to the claw tip for use in making longer wounds (fig 1 C, D):

" Interestingly, the magnitude of force capable of being produced at the claw tip was relatively small, arguing against it being used for transmitting a large proportion of body weight into a substrate. Collectively, the results best support a grasping function for digit II (e.g., restraint of small-bodied prey), although other behaviours involving flexed hindlimbs cannot be excluded."

At first read, this sounds like a rebuttal to our suggestion that a dromaeosaur might stand on top of its prey to hold it down using its body weight (as modern birds of prey do; Fowler et al., 2009; 2011). I suppose that our papers are perhaps not clear enough when we state that the inside toe (D-II) is better placed to exert body weight of the animal on to prey. As I have mentioned in the comments on the manuscript DOC, when a modern bird of prey is standing on top of its victim, its body weight pins down the victim - the body weight is not transferred only through the claws, but through the toes and metatarsus too (especially when the metatarsals contact the prey. This was well illustrated for birds of prey by Goslow, 1972). However, I understand that the current manuscript abstract is only talking about the D-II claw. I think this could be made more clear, because I don't think the findings of this analysis refute our suggestion that bodyweight helps pin down the prey, only that the bodyweight is not entirely/mainly applied through the D-II ungual.

There needs to be a definition of what is meant by prey size categories - we do this in Fowler et al. (2009), and extend it in Fowler et al. (2011). We define prey size relatively: if prey can be contained within the foot of the predator, then it is a "small" prey item. If the prey cannot be contained within a foot, but can be held down by the body weight of the predator (if the predator stands on top of it) then it is "large" prey. If the prey is so large that it cannot be held down by the predator, then we call it "very large" prey.

Defining prey size is quite important as in the conclusions and abstract it is stated " Collectively, the results best support a grasping function for digit II (e.g., restraint of small-bodied prey" - but I don't think "small prey" is meant in the same way that we would define it (Fig 1, F, would be considered "large prey" in our definition).

Finally, I agree that the claws are probably not exerting major force, but this does not mean that they are not useful for subduing very large prey (under our definition, see above). Imagine that a person is clinging to the back of an angry cow which is shaking about violently (as you might see in some western agricultural shows) - the person uses their fingertips to hang on for grip, they don't exert much force on the fingertips, and certainly don't hang all their weight on their fingertips (as a rock climber might), but they are maintaining grip. In this situation the person is acting similar to what we describe as "prey riding" - a behavior observed in birds of prey that attack what we categorise as "very large prey" (see Fowler et al., 2009; 2011; and above). Thus even without exerting strong force, claws help maintain grip on a prey items of "very large" size (indeed, all sizes).

Reviewer 2 ·

Basic reporting

The article is well structured, but I have some issues with the figures - I appreciate these models output a great deal of information, but I am not sure that Figures 4-6 in their current form are the best way of presenting it.

The raw data have been shared, but not all of it is in an immediately accessible format. The code itself can be opened and viewed in any good text editor, but the associated data files cannot. Would it be possible to output the modelling data as .csv files rather than Matlab data objects?

Experimental design

Regarding the values for pennation angles, how was the initial range of values chosen? As part of your sensitivity analyses you used a more restricted range of values which was "comparable to the range observed in the digital flexor and extensor muscles of extant archosaurs". Could you please explain and justify then the range chosen in the initial system configuration which includes values well outside those seen in extant archosaurs? Why did you not use the values from extant taxa to begin with and then extend the range in the sensitivity analyses? As far as I can tell, all the other assumptions and initial values (flexor and extensor lengths, muscle volumes etc), appear well explained and justified so why pennation angles aren't is puzzling.

Validity of the findings

Whilst I believe the overall findings are sound, the methodological questions I raised in the previous section mean I do have some differences in interpretation. I think the author should base their interpretations around results of the models with realistic pennation angles akin to those seen in extant archosaurs, and discuss what is currently presented as their main findings as a sensitivity analysis to test the effect of pennation angles outside of what we know from extant taxa. This would change the results somewhat – when restricted to more “realistic” pennation angles claw forces are substantially higher at extended angles and erect postures. However, at the more flexed angles the author suggests the claws operate, force are still highest with a crouched posture.

Additional comments

I think this is a valuable contribution to the literature, using biomechanical models to explore possible functions of the famous "killing claw" of Deinonychus and other raptors. The author explores a wide range of parameter-space and sensitivity analyses to identify those features, e.g. limb posture, which consistently correlate to relevant biomechanical measures of claw function. I do have certain reservations about the paper in its current form, which I have outline above, but addressing these should be feasible as (I think) they only require re-interpretation or re-analysis of the model outputs presented in the paper and supplementary information, rather than a complete rerun of the whole modelling process.

In addition to my comments above, I feel the graphs in Figures 4-6 could be improved - showing every single one of the 108 curves leads to very busy plots. I think plotting averaged curves with confidence intervals or min/max ranges for individual conditions would be better (i.e. in Figure 4A have three curves with confidence intervals, one for each posture etc.). Or perhaps running principal components analysis on the force-angle data (where the input into the PCA would be the value of force at each MTP angle)? This would also provide a means of comparing both curve magnitude and shape depending on whether you scaled the values or not.

I also wonder whether it is necessary to display the results for the full range (-65 to 60) of MTP angles? I assume they are mostly for illustrative purposes, but in Figure 1 the only behaviours shown in which the claws might actually reach substantial flexion are digging or holding small prey. All other behaviours show the claw either only slightly flexed, or extended. If posture and claw angle could be somehow codified for these different behaviours (perhaps with an added table?) I think this would make the links between the modelling results and conclusions about behaviour easier for the reader to see.

I appreciate the work the author has done in this paper and I believe it should be published. Whilst my comments should hopefully be straightforward to address I would prefer to re-evaluate any revised version, and so in-line with PeerJ reviewer guidelines I must return a verdict of major revisions.

---

## Round 0.2 · accepted · Accept

Your revisions were re-reviewed by the reviewer who raised some substantial comments in the first round of review. The reviewer was pleased to see that all their comments had been fully addressed in this revision, making the text much stronger and more accessible for the reader. As such, the reviewer is satisfied with the changes and the latest version of the text, recommending it for publication as is. I look forward to seeing your paper in press.

Reviewer 2 ·

Basic reporting

No comments

Experimental design

No comments

Validity of the findings

No comments

Additional comments

I have reviewed the revised manuscript and I think it is greatly improved. I appreciate the work the author has done on this new version in response to my comments and those of my fellow reviewers. The author has managed to sufficiently address all of my points from my review of their original submission. I am particularly thankful for the additional clarification and discussion around pennation angles and general variable choice in models such as this, as well as the new table (Table 2). I think the manuscript should be published in its current form.